# Curcumin-Loaded Platelet Membrane Bioinspired Chitosan-Modified Liposome for Effective Cancer Therapy

**DOI:** 10.3390/pharmaceutics15020631

**Published:** 2023-02-13

**Authors:** Shengli Wan, Qingze Fan, Yuesong Wu, Jingqing Zhang, Gan Qiao, Nan Jiang, Jie Yang, Yuanzhi Liu, Jingyan Li, Sawitree Chiampanichayakul, Singkome Tima, Fei Tong, Songyot Anuchapreeda, Jianming Wu

**Affiliations:** 1Division of Clinical Microscopy, Department of Medical Technology, Faculty of Associated Medical Sciences, Chiang Mai University, Chiang Mai 50200, Thailand; 2Department of Pharmacy, The Affiliated Hospital of Southwest Medical University, Luzhou 646000, China; 3School of Basic Medical Sciences, Southwest Medical University, Luzhou 646000, China; 4School of Pharmacy, Southwest Medical University, Luzhou 646000, China; 5Key Laboratory of Medical Electrophysiology of Ministry of Education of China, School of Pharmacy, Southwest Medical University, Luzhou 646000, China; 6Chongqing Research Center for Pharmaceutical Engineering, Chongqing Medical University, Chongqing 400016, China; 7Center for Research and Development of Natural Products for Health, Chiang Mai University, Chiang Mai 50200, Thailand

**Keywords:** biomimetic camouflage, platelet membrane, liposome, tumor-targeted delivery, curcumin

## Abstract

Cancer is a serious threat to human health, and chemotherapy for cancer is limited by severe side effects. Curcumin (CUR) is a commonly used natural product for antitumor treatment without safety concerns. However, low bioavailability and poor tumor accumulation are great obstacles for its clinical application. Our previous research has demonstrated that platelet membrane-camouflaged nanoparticles can efficiently ameliorate the in vivo kinetic characteristics and enhance the tumor affinity of payloads. Nevertheless, the antitumor efficiency of this formulation still needs to be thoroughly investigated, and its drug release behavior is limited. Herein, CUR-loaded platelet membrane bioinspired chitosan-modified liposome (PCLP-CUR) was constructed to improve CUR release. PCLP-CUR was shown to have long retention time, improved bioavailability, strong tumor targeting capacity and effective cellular uptake. The incorporation of chitosan enabled PCLP-CUR to release cargoes quickly under mild acidic tumor conditions, leading to more complete drug release and favoring subsequent treatment. Both in vitro and in vivo investigations showed that PCLP-CUR could significantly enhance the anticancer efficacy of CUR with minimal side effects through biomimetic membrane and chitosan modification. In summary, this developed delivery system can provide a promising strategy for tumor-targeting therapy and phytochemical delivery.

## 1. Introduction

Tumors are a severe threat to human health [1]. Systemic chemotherapy is the main option for cancer patients. Nevertheless, the undesirable side effects and toxicity of chemotherapeutic drugs have given rise to damage to patients [2]. Nutraceuticals are intensively investigated in cancer and chronic diseases, and these natural products offer new possibilities for tumor therapies to overcome the shortcomings of traditional chemotherapeutic treatment [3,4,5]. Curcumin (CUR), a curcuminoid extracted from the rhizomes of *Curcuma longa*, has several advantages, such as strong anticancer effects and outstanding safety [6,7,8,9]. Nevertheless, CUR is a hydrophobic molecule with low bioavailability [10]. Meanwhile, rapid blood clearance and limited tumor accumulation have hindered the clinical application of natural compounds, including CUR.

Nowadays, nanonutraceutical formulations represent a valuable strategy used in managing health conditions. In particular, nanoscale liposomes have already drawn interest in cancer therapy for the enhancement of physicochemical properties for traditional therapeutic agents and preferential accumulation in tumor tissues via the enhanced permeability and retention (EPR) effect [11,12,13,14]. However, there are still often problems of limited cellular uptake and inadequate drug release for nanoformulations. Chitosan, a positively charged polysaccharide, has been widely employed in drug delivery field due to its low toxicity and biodegradable property. Tumors display a lower extracellular pH than normal tissues, as well as in their intracellular lysosomes (pH~5.0), thus providing an ideal platform for designing stimulus-responsive drug release systems [15]. Chitosan-based delivery systems can achieve pH-responsive drug release via swelling and protonation, leading to more complete drug release in acidic environments and favoring subsequent anticancer therapy [15,16].

However, simple surface functionalization strategies cannot precisely mimic the complicated interfaces present in nature and will eventually be seen as foreign bodies and induce immune responses [17,18]. Recently, cell membrane biomimetic nanoplatforms have attracted increasing attention in biomedicine applications due to their ingenious functions, such as superior biocompatibility, low immunogenicity and active cancer targeting ability. Membranes extracted from platelets have been reported to be sufficiently efficient in protecting nanoparticles from immune activation and rapid clearance due to the presence of the “maker-of-self” protein (integrin-associated protein, CD47) on their surface [19]. CD47 can protect nanoparticles from phagocytosis by macrophages through interactions with signal regulatory protein α and sending out a “don’t eat me” signal to macrophages [19,20]. In addition, platelets show strong affinity for a variety of cancer cells [21,22,23,24,25]. Hepatocellular carcinoma HepG2 cells, which are highly CD44-expressing cells [26], have been reported to aggregate platelets irreversibly [27]. In our previous study, platelet membrane biomimetic nanoparticles showed enhanced cytotoxicity against liver cancer cells [28]. So far, liver cancer is one of the most fatal cancers with worldwide prevalence [29]. In this study, liver cancer was used as a disease model to explore a more effective cancer treatment. To date, reports on the delivery of natural antitumor drugs by platelet membrane biomimetic nanoparticles are still limited.

In our work, a novel CUR-loaded platelet membrane bioinspired chitosan-modified liposome (PCLP-CUR) was developed to improve the release, pharmacokinetic characteristics, tumor targeting and anticancer effect of CUR. CUR-loaded chitosan-modified liposome (CLP-CUR) was first prepared, followed by cloaking with platelet membrane to form PCLP-CUR (Figure 1A). As illustrated in Figure 1B, PCLP-CUR displayed increased retention in the circulation and improved bioavailability through platelet membrane camouflage. Upon accumulation in tumors via the EPR effect and specific intracellular uptake, the incorporation of chitosan enabled PCLP-CUR to respond to the intracellular pH of cancer cells and induce the efficient release of cytotoxic drugs. Under the combined effect of platelet membrane cloaking and chitosan modification, PCLP-CUR exhibited strong antitumor efficiency.

## 2. Materials and Methods

### 2.1. Materials

Cholesterol was obtained from Sigma-Aldrich Chemical Co. (St. Louis, MO, USA). D-α-tocopheryl polyethylene glycol 1000 succinate (TPGS), hyaluronic acid (HA) and egg yolk lecithin were purchased from Meilun Biotechnology Co., Ltd. (Dalian, China). Chitosan hydrochloride was obtained from Golden-Shell Pharmaceutical Co., Ltd. (Taizhou, China). CUR was purchased from Yuanye Biotechnology Co., Ltd. (Shanghai, China). Polycarbonate porous membrane and Avanti mini extruder were purchased from Avanti Polar Lipids (Alabaster, AL, USA). Phosphate buffered saline (PBS) was obtained from Solarbio Science & Technology Co., Ltd. (Beijing, China). Fetal bovine serum (FBS) and Dulbecco’s modified Eagle medium (DMEM) were obtained from Gibco Invitrogen Corp. (Waltham, MA, USA). In addition, 1,1′-Dioctadecyl-3,3,3′,3′-tetramethylindodicarbocyanine, 4-chlorobenzenesulfonate salt (DiD), 3,3′-dioctadecyloxacarbocyanine perchlorate (DiO) and 1,1′-dioctadecyl-3,3,3′,3′-tetramethylindotricarbocyanine iodide (DiR) were obtained from Lablead Biotech Co. Ltd. (Beijing, China). Hoechst 33342 was purchased from Cell Signaling Technology Inc. (Danvers, MA, USA). Mouse anti-P-Selectin antibody and rabbit anti-CD41 antibody were purchased from Proteintech Group, Inc. (Wuhan, China). Rabbit anti-CD47 antibody was purchased from ABclonal Technology (Wuhan, China). Bovine serum albumin (BSA), 4′,6-diamidino-2-phenylindole (DAPI) and rabbit anti-CD44 antibody were purchased from Servicebio Co., Ltd. (Wuhan, China). A bicinchoninic acid (BCA) protein assay kit was purchased from Bio-Rad Laboratories, Inc. (Hercules, CA, USA).

Male BALB/c nude mice (14–16 g) were obtained from Huafukang Bioscience Co., Inc. (Beijing, China), and female Sprague Dawley (SD) rats (160 ± 20 g) were supplied by the Laboratory Animal Center at Southwest Medical University (Luzhou, China). All animals were housed under specific pathogen-free (SPF) conditions.

The hepatocellular carcinoma cell line (HepG2) and mouse macrophage cell line (RAW264.7) were obtained from American Type Culture Collection (ATCC). Human umbilical vein endothelial cells (HUVECs) were a generous gift from the Laboratory for Cardiovascular Pharmacology of Department of Pharmacology, School of Pharmacy, Southwest Medical University, China. All cells were cultured in DMEM supplemented with 10% FBS under a 5% CO_2_ atmosphere at 37 °C.

### 2.2. Fabrication of CLP-CUR

The CLP-CUR was prepared by our previously reported thin film dispersion method with slight modifications [28,30]. In brief, egg yolk lecithin, cholesterol, TPGS and CUR (8 mg) were dissolved in 30 mL of dichloromethane. The solvent was evaporated to form a dry lipid film. Next, the film was hydrated for 2 h with 10 mL of PBS to form an aqueous suspension; thus, CUR-loaded liposome (LP-CUR) was prepared. Then, chitosan solution (0.2%, weight/volume) was further dropwise added to this suspension under magnetic stirring for 60 min at 25 °C to prepare CLP-CUR.

### 2.3. Fabrication of PCLP-CUR

The platelet membranes were extracted according to a previous reported repeated freeze-thaw method with slight modification [31]. Briefly, platelets from whole blood samples of SD rats were isolated by differential centrifugation. The platelets were frozen at −80 °C and thawed at room temperature, and this process was repeated three times. Then, the membrane was collected by centrifugation at 20,000× *g* for 40 min. After washing with PBS buffer containing protease inhibitors, the membranes were then sonicated for 5 min to fabricate platelet membrane nanovesicles (PNVs). To prepare PCLP-CUR, the obtained PNV was incubated with CLP-CUR for 30 min, and the resultant suspensions were sonicated for 5 min and then extruded sequentially through polycarbonate membranes with an Avanti Polar Lipids mini extruder [31,32,33]. CUR-loaded platelet membrane biomimetic liposome (PLP-CUR) was prepared by the same method without the addition of chitosan. To characterize the morphology, the liposomes were examined using a transmission electron microscope (HT7700, Hitachi, Tokyo, Japan). The hydrodynamic size and Zeta potential of liposomes were measured using a Zetasizer (Nano ZS, Malvern, UK).

### 2.4. Colocalization of Platelet Membrane and CLP on PCLP

Confocal microscopy assay was carried out to confirm membrane integration of PNV and CLP. CLP and PNV were, respectively, marked with the red fluorescent dye DiD and green, fluorescent dye DiO [21,31]. PCLP was prepared with the resultant DiO-labeled PNV and DiD-labeled CLP and then visualized using confocal laser scanning microscopy (TCS SP8, Leica, Wetzlar, Germany).

### 2.5. Analysis of Platelet Membrane Proteins

Protein profiles of PLP-CUR, PCLP-CUR and PNV were analyzed by sodium dodecyl sulfate-polyacrylamide gel electrophoresis (SDS-PAGE) and Western blot assay. These vesicles were normalized to an equivalent protein concentration with a BCA assay kit. All samples were prepared in loading buffer, boiled for 5 min and then separated on a 10% SDS-PAGE gel. Coomassie brilliant blue R-250 was used to stain the gel for observation. For Western blot analysis, the proteins on the gel were transferred to a polyvinylidene difluoride membrane. Following blocking with 5% nonfat milk, the membranes were probed with primary antibodies against P-selectin, CD47 and CD41 at 4 °C overnight, followed by appropriate horseradish peroxidase-conjugated secondary antibodies incubation, and then detected using an imaging system (BOX EF2, Syngene G, Cambridge, UK).

### 2.6. Encapsulation Efficiency and Stability

Freshly prepared PCLP-CUR was centrifuged at 20,000× *g* for 40 min to separate the free drug from the encapsulated one. The supernatant was collected, and a UV spectrophotometer at 426 nm was used to quantify the content of free CUR. The amount of encapsulated CUR was calculated by subtracting the unencapsulated CUR from the total input. The encapsulation efficiency was calculated using Equation (1):(1)$$\mathrm{Encapsulation}\;\mathrm{efficiency}\;(\%)=\frac{\mathrm{Mass}\;\mathrm{of}\;\mathrm{CUR}\;\mathrm{encapsulated}\;\mathrm{in}\;\mathrm{PCLP}}{\mathrm{Total}\;\mathrm{mass}\;\mathrm{of}\;\mathrm{CUR}}100\%$$

For the stability study, the PCLP-CUR was stored at 4 °C, and encapsulation efficiency was measured periodically over the course of 10 days.

### 2.7. In Vitro Drug Release

The CUR release capacity from PCLP-CUR or PLP-CUR was studied by a dynamic dialysis method as previously reported [28]. Briefly, 1 mL of PCLP-CUR or PLP-CUR suspensions was added into a dialysis bag (molecular weight cutoff, 7000 Da) and dialyzed against 100 mL of PBS release medium (pH 5.0, 6.5, or 7.4) containing 0.5% sodium dodecyl sulfate and 20% ethanol. The release experiment was performed at 37 °C in an incubator with shaking (100 rpm). At different time points, the external medium (1 mL) was collected and replaced with 1 mL of fresh release media. Released CUR was quantified by a UV spectrophotometer at 426 nm.

### 2.8. Cytocompatibility

An MTT assay was performed to evaluate the in vitro cytocompatibility of blank PCLP. HUVECs and HepG2 cells were cultured in 96-well plates at a density of 5 × 10^3^ cells/well, following treatment with 100 μL of PCLP (0, 25, 50, 200, 400 or 800 μg/mL) for 48 h. At the end of the experiment, cells were incubated with MTT dye solution (0.5 mg/mL) at 37 °C for 4 h. The formazan product was solubilized with 150 μL of DMSO and then measured at 570 nm using a microplate reader (VLBL00D0, Thermo Fisher, Waltham, MA, USA).

### 2.9. Cytotoxicity Assay

The cytotoxicity of CUR, LP-CUR, CLP-CUR, PLP-CUR or PCLP-CUR against HepG2 cells was evaluated with an MTT assay. HepG2 cells were cultured in 96-well plates at a density of 5 × 10^3^ cells/well and then treated with CUR or CUR-loaded liposomes (20 µM equivalent dose of CUR) for 48 h. After treatment, cells were determined by the MTT method as described above.

### 2.10. Cellular Uptake Evaluation

The cellular uptake of different formulations was measured by confocal microscopy. HepG2 cells or RAW264.7 cells (200 µL) were seeded into confocal dishes at a density of 5 × 10^4^ cells per well and cultured overnight. LP-DiD, CLP-DiD or PCLP-DiD (200 µL) were then added and further incubated for 2 h. Then, the cells were washed three times with PBS and fixed with 4% paraformaldehyde (PFA), and the nuclei were stained by Hoechst 33342. After being washed with PBS, the cells were imaged with confocal laser scanning microscopy. To explore the CD44 receptor-mediated cellular uptake of PCLP in HepG2 cells, HepG2 cells were pretreated with free HA (10 mg/mL) for 3 h [34]. After removing free HA, the cells were treated with PCLP-DiD for another 2 h. Then, the cells were processed and examined as described above.

### 2.11. Pharmacokinetics of PCLP-CUR in Rats

Fifteen SD rats (five rats per group) were intravenously injected with CUR, LP-CUR or PCLP-CUR at the same CUR dose (8 mg/kg). Blood samples were collected from the posterior orbital venous plexus after injection at predetermined time points and subjected to centrifugation. The supernatant was harvested to obtain the plasma samples. CUR concentrations in the plasma samples were determined by our previously reported HPLC method [28]. In brief, the plasma samples were added to a mixture of methanol and ethyl acetate (1: 9, *v*/*v*) and then vortexed for 5 min. After centrifugation (12,000 rpm for 10 min), the supernatant was collected and then subjected to evaporate under nitrogen. The residue was redissolved and centrifuged at 12,000 rpm for 10 min. The supernatant (20 μL) was injected into an HPLC system (Agilent-1260, Santa Clara, CA, USA) for analysis at 426 nm with a Kromasil 100-5-C18, 4.6 × 250 mm column. The mobile phase was composed of acetonitrile and 5% acetic acid (50:50, *v*/*v*) at a flow rate of 1 mL/min. The major pharmacokinetic parameters were calculated with a DAS 3.0 software (Mathematical Pharmacology Professional Committee of China, Shanghai, China).

### 2.12. Biodistribution

Fluorescence imaging was applied to investigate the in vivo tumor targeting ability and biodistribution of PCLP. PCLP and LP encapsulated with the fluorescent dye DiR were prepared by the methods described above, but DiR was added instead of CUR. HepG2 cells (4 × 10^6^) in PBS (100 μL) were injected subcutaneously into the left flank of BALB/c nude mice to construct the HepG2 tumor-bearing mouse model. When the tumor volume reached an average size of ~150 mm^3^, PCLP-DiR and LP-DiR (500 μg/kg equivalent dose of DiR) were intravenously injected into the HepG2 tumor-bearing nude mice. Fluorescence imaging at various time intervals after administration was performed on a multimodal animal imaging system (BLT Aniview 100, Biolight Biotechnology, Guangzhou, China). At 24 h postinjection, the mice were sacrificed, and the tumors and major organs, including the heart, lung, spleen, liver, and kidney were collected for ex vivo imaging and quantification.

### 2.13. Immunofluorescence Assay

To further highlight the targeted effect of PCLP, HepG2 tumor-bearing nude mice were injected with fluorescent dye Cy5-labeled liposomes via the tail vein. The tumors from mice after treatment with PCLP-Cy5, CLP-Cy5 and LP-Cy5 for 24 h were harvested. Afterward, the tumor tissues were whole-mounted and cut into 4 μm slices. After blocking with 5% BSA, these tumor sections were incubated with primary antibody against CD44 at 4 °C overnight followed by incubation with DyLight 488-conjugated goat anti-rabbit IgG at 37 °C for 50 min. Samples were stained with DAPI to detect nuclei as a counterstain.

### 2.14. In Vivo Antitumor Efficacy

When the tumor volumes grew to an average size of ~100 mm^3^, the HepG2 tumor-bearing nude mice were randomly divided into seven groups and injected intravenously with PBS (pH 7.4), free CUR, PCLP, LP-CUR, CLP-CUR, PLP-CUR or PCLP-CUR (10 mg/kg equivalent dose of CUR) once every 2 days for a total of 7 administrations. Tumor size [volume = (length × width^2^)/2] and body weight were measured over time. After the treatment, blood samples were collected and centrifuged at 3500 rpm for 15 min at 4 °C to obtain serum. The concentrations of creatine kinase (CK), lactate dehydrogenase (LDH), alanine aminotransferase (ALT), aspartate aminotransferase (AST), creatinine (CREA) and urea (UREA) were analyzed using an automatic biochemistry analyzer (ADVIA 2400, Siemens, Munich, Germany). The tumor tissues were further detected by TUNEL assay to examine tumor cell apoptosis and Ki-67 staining to evaluate tumor cell proliferation. The major organs were removed for histological assessment by H & E staining.

### 2.15. In Vivo Safety

Healthy BALB/c nude mice were injected intravenously with PBS (pH 7.4), CUR, PCLP and PCLP-CUR every two days for a total of seven administrations. Serum samples were collected from the mice following the last treatment. The levels of CK, LDH, ALT, AST, CREA and UREA were detected in the serum samples. The main organs of the mice were also collected for H & E staining.

### 2.16. Statistical Analysis

Data presented in this work were expressed as the means ± SDs. Data analysis was conducted using the software GraphPad Prism 8. Both one-way ANOVA and Student’s *t*-test were utilized for statistical evaluation. *p* < 0.05 indicated a significant difference.

## 3. Results

### 3.1. Fabrication and Characterization of PCLP-CUR

Transmission electron microscope images showed that all nanoparticles were spherical (Figure 1A). PCLP-CUR had an average hydrodynamic diameter of 162.8 nm and possessed high encapsulation efficiency with a value of 91.69%. This developed nanoparticle showed few changes in encapsulation efficiency over time (Figure 1B), indicating good stability. As shown in Figure 1C, the Zeta potential of CLP-CUR was reversed from −43.7 mV (zeta potential of LP-CUR) to 42.0 mV after chitosan modification, and the electrical transition induced by chitosan cloaking indicated the successful preparation of CLP-CUR. The Zeta potential of PLP-CUR (−38.1 mV) was quite similar to that of the purified platelet membranes (−32.8 mV). In addition, the Zeta potential of PCLP-CUR after fusion with the platelet membrane decreased to 22.2 mV. This difference in the Zeta potential between PCLP-CUR and CLP-CUR suggested the successful fabrication of PCLP-CUR. In this study, PCLP-CUR was a platelet membrane-lipid hybrid drug carrier; thus, its Zeta potential was not very close to that of the platelet membrane, which was similar to the previously reported platesomes [31]. To further confirm the successful fusion between platelet membrane and lipid membrane on PCLP, confocal microscopy analysis was carried out. When PCLP was prepared using DiO-labeled PNV and DiD-labeled CLP, the green and red fluorescence completely merged to display yellow fluorescence, suggesting the integration of the PNV and lipid membrane (Figure 1D). Furthermore, SDS-PAGE result certified that biomimetic liposomes had similar protein profiles to platelet membrane (Figure 1E). The Western blot result showed that bioinspired liposomes retained the characteristic proteins on the platelet membrane, including the platelet-specific CD41, immunomodulatory CD47 and cancer-targeted P-selectin (Figure 1F). In conclusion, PCLP-CUR was successfully prepared with a protein environment quite similar to the platelet membrane source.

### 3.2. Drug Release

Drug release from PCLP-CUR or PLP-CUR was measured at pH 7.4 (simulating the normal physiological environment), pH 6.5 and pH 5.0 (simulating the acidic microenvironment of the tumor) [35]. A typical biphasic pattern with an initial quick-release phase followed by a slow sustained-release phase was observed in both the PCLP-CUR group and PLP-CUR group (Figure 1G,H). After dialysis for 24 h, the release rates of CUR from PCLP-CUR were approximately 60% at pH 5.0, 49% at pH 6.5 and only 34% at pH 7.4 (Figure 1G). Drug release from PCLP-CUR was much faster in the acidic environment than that at pH 7.4. In contrast, the drug release profiles of PLP-CUR, the formulation without chitosan modification, were similar at different pH conditions (Figure 1H).

### 3.3. Efficient In Vitro Delivery of PCLP

Cell membrane biomimetic nanoparticles can prolong the in vivo circulation time through immune escape [24]. To demonstrate the immune escape ability of PCLP, the phagocytosis of RAW264.7 macrophages was evaluated. RAW264.7 cells treated with LP-DiD or CLP-DiD showed higher red fluorescence intensities than those treated with PCLP-DiD (Figure 2A), suggesting that the prepared bioinspired liposome could escape uptake by macrophages. The uptake of PCLP-DiD in CD44-overexpressing HepG2 cells was also examined. The fluorescence signals in HepG2 cells treated with PCLP-DiD were stronger than those of the synthetic liposome groups (Figure 2B), indicating that PCLP-DiD was more efficiently taken up by HepG2 cells. These results confirmed that PCLP could successfully escape uptake by macrophages and drive cargo into HepG2 cells efficiently. To further explore the endocytosis mechanism of cellular uptake, an experiment of competitive inhibition was performed in which HepG2 cells were pretreated with free HA. HA is a strong-affinity ligand of the CD44 receptor [34]. Confocal laser scanning microscope images (Figure 2B) showed that the fluorescence intensity of PCLP in free HA preincubated HepG2 cells was reduced. This result indicates that the CD44 receptor was involved in the recognition process of the developed biomimetic liposome and HepG2 cell.

### 3.4. Pharmacokinetics

To highlight the prolonged circulation time and enhanced bioavailability of PCLP-CUR, pharmacokinetics were then investigated. The plasma drug concentration-time curve was shown in Figure 3A. The area under the concentration-time curve (*AUC*) of PCLP-CUR was 336.99 μg/L·h, which was approximately 2.51-fold that of CUR and 1.44-fold that of LP-CUR (Figure 3B). This result indicates that the bioavailability of PCLP-CUR was significantly enhanced compared with that of free CUR and LP-CUR. The mean residence time (*MRT*) of PCLP-CUR was 3.60 times that of CUR and 2.15 times that of LP-CUR (Figure 3C), while both CUR and LP-CUR had a significantly higher clearance rate (*Cl*) than PCLP-CUR (Figure 3D). PCLP-CUR with longer *MRT* and lower *Cl* values maintained a prolonged circulation time compared to LP-CUR and free CUR.

### 3.5. In Vivo Tumor Targeting Capability of PCLP

Afterward, the in vivo tumor targeting effect of PCLP was evaluated. First, the biodistribution of PCLP in HepG2 tumor-bearing nude mice was investigated. As shown in Figure 4A, at 24 h after intravenous administration, LP-DiR was found to moderately concentrate at the tumor sites, which was likely attributed to the EPR effect of liposome. In contrast, the fluorescence intensity of PCLP-DiR in tumor tissues was higher and lasted longer than that of the LP-DiR group, suggesting that the developed bioinspired liposome had a superior targeting capacity. The ex vivo images of excised organs at 24 h after treatment also confirmed this result. As shown in Figure 4B,C, a higher DiR signal was observed in the tumor sites of PCLP-DiR group than that of LP-DiR group. These results were in good accordance with the in vitro cellular uptake. The sustained high-dose retention in tumor tissues makes PCLP a promising delivery system for anticancer agents.

Immunofluorescence assay was applied to further highlight the targeting capacity of PCLP. As shown in Appendix A, strong green fluorescence could be detected in all the tumor sections, suggesting the expression of CD44 on the HepG2 cells. Strong red fluorescence signals of PCLP-Cy5 could be detected in tumor tissues, while the tumor tissues harvested from mice treated with synthetic liposomes showed weak red fluorescence intensities. The result demonstrated that platelet membrane could drive the developed biomimetic liposomes to specifically bind to cancer cells. Based on these results, it could be concluded that PCLP displayed favorable targeting capacity.

### 3.6. Enhanced Therapeutic Efficacy of PCLP-CUR

Encouraged by the good targeting ability, the therapeutic effect of PCLP-CUR was examined. As shown in Figure 5A, PCLP-CUR showed a significantly stronger tumor cell-killing effect than CUR. Upon chitosan modification, CLP-CUR exhibited higher cytotoxicity than LP-CUR. In addition, compared with CLP-CUR, PCLP-CUR showed a notably higher cytotoxicity after cloaking with the platelet membrane. The in vivo therapeutic efficacy of PCLP-CUR was subsequently evaluated. Tumor-bearing BALB/c nude mice were injected with various formulations via the tail vein once every two days for a total of seven injections (Figure 5B). As shown in Figure 5C,D, when compared with control group and blank PCLP, both CUR and LP-CUR exhibited a moderate effect on the inhibition of tumor growth, while an observably enhanced tumor inhibition effect was found in PCLP-CUR group. The therapeutic effect was further confirmed by Ki-67 staining and TUNEL staining. Ki-67, as a marker for cell proliferation, is widely used to assess the antitumor efficacy on tumor cell proliferation. TUNEL assay is generally used to detect the apoptotic programmed cell death. As shown in Figure 5E, numerous apoptotic cells (TUNEL-positive cells) were observed in PCLP-CUR group, and the number of proliferating cells (Ki-67-positive cells) decreased. Both in vitro and in vivo results showed that PCLP-CUR could significantly enhance the anticancer effect of CUR, which was likely ascribed to the chitosan modification and platelet membrane cloaking. The favorable anticancer performance of PCLP-CUR indicates the great potential of this bioinspired nanocarrier for targeted cancer therapy.

### 3.7. Biosafety

To further evaluate the potential of the prepared biomimetic liposome in biomedical applications, the biosafety was examined. There was no obvious cytotoxicity after blank PCLP incubation with HUVECs or HepG2 cells even when the concentration of blank vesicles reached 800 μg/mL (Figure 6A,B), indicating that PCLP had good cytocompatibility. The safety of the developed formulations in vivo was also explored. HepG2 tumor-bearing mice were treated with different formulations, and a continuous body weight increase was observed in all treated mice without any discomfort or death (Figure 6C), certifying that all the formulations were quite safe. Furthermore, the H & E staining of tissues showed no obvious damage to the heart, liver, spleen, lung or kidney in all groups (Figure 6D). The serum biochemistry analysis showed that the liver function indexes (ALT and AST), heart function indexes (CK and LDH) and kidney function indexes (CREA and UREA) were very similar to those of the PBS group (Figure 6E–G), suggesting no obvious hepatic, cardiac or kidney disorders in the treatment.

The in vivo toxicity in healthy nude mice was also investigated. H & E staining did not show any apparent damage in any of the major organs (Appendix A). In addition, PCLP-CUR did not cause any obvious changes in biomarkers of organ function versus the PBS group (Appendix A). These results demonstrate the PCLP-CUR had good biosafety.

## 4. Discussion

Natural products are a sustainable bioresource with the potential to treat cancer. Many efforts, such as nanotechnology, the integrated study approach and network, have been devoted to developing effective natural medicines [4,36]. However, the delivery of natural anticancer drugs faces great challenges due to limited in vivo kinetic behaviors and poor tumor accumulation. To improve their bioavailability and antitumor effect, we prepared novel biomimetic polysaccharide-cloaked lipidic nanocarriers (PCLP-CUR) to deliver the natural polyphenol CUR. This developed nanoparticle is safe and has high efficacy.

The micromorphology, size, Zeta potential and protein environment suggested the successful fabrication of PCLP-CUR. Chitosan modification ameliorated drug release behavior [37]. According to the drug release rate, the drug release from PCLP-CUR was more rapid in acidic environment than at pH 7.4 (Figure 1G), indicating the pH-responsive drug release ability of PCLP-CUR. The mechanism might be attributed to the protonated amines of chitosan, which can induce the swelling of PCLP-CUR; thus, the CUR molecules could easily diffuse out from the liposomes [15,16]. Because the anticancer drug release was much faster and greater under acidic conditions than under normal physiological conditions, the side effects of drug could be significantly reduced in normal cells [15,38]. Thus, the newly prepared PCLP-CUR with pH-responsivity could achieve tumor treatment and reduce the adverse effect of therapeutic agents.

The results of cellular uptake demonstrated that PCLP could escape uptake by macrophages (Figure 2A), presumably the CD47 protein on the platelet membrane sending “don’t eat me” signals to macrophages [39,40]. Phagocytosis of macrophage cells is the primary cause of the rapid blood clearance of drug carriers [41]. The prolonged *MRT* values and decreased *Cl* suggested the in vivo long circulation of PCLP. The ameliorated in vivo kinetic characteristics should be ascribed to platelet membrane camouflage.

Both in vitro and in vivo assays confirmed that PCLP had a good tumor targeting capacity. As shown in Figure 2B, compared with synthetic liposomes, PCLP was more efficiently taken up by HepG2 cells. Moreover, the uptake of PCLP in free HA (a ligand of CD44) preincubated HepG2 cells was reduced. As described in previous reports, P-selectin has a high affinity for the CD44 receptor [31,41]. The enhanced cellular uptake of PCLP was likely attributed to the interaction between the CD44 receptor of HepG2 cell and P-selectin on the platelet membrane. In vivo tumor targeting evaluation confirmed that the accumulation of PCLP in tumor tissues was greater than that of synthetic liposomes, presumably through the platelet membrane cloak [39].

The in vivo anticancer study showed that the tumor size of HepG2 tumor-bearing mice treated with PCLP-CUR was obviously lower than those of the control group and free drug group. The in vivo results of PCLP-CUR were in line with the in vitro antitumor effects. The high anticancer efficiency of PCLP-CUR might be due to the following reasons: the entrapment of CUR inside of the PCLP-CUR improved its in vivo retention time, bioavailability and tumor specificity, which allowed more drug accumulation at the cancerous site; platelet membrane camouflage contributed to the increasing uptake of CUR by cancer cells through interaction between P-selectin and CD44 [42]; and the modification of chitosan accelerated drug release at the tumor regions [35]. Weight change, organ H & E staining and serum biochemistry analysis certified that the preliminary safety of PCLP-CUR was good. The cytocompatibility assay also confirmed the high biocompatibility of the developed nanocarrier. This satisfactory safety may be related to the materials used in this study, such as natural polysaccharides and biocompatible cell membranes.

Based on these advantages, PCLP-CUR represents a promising targeted cancer treatment. However, the safety and antitumor efficiency of emerging biomimetic nanoformulations have not been fully demonstrated in humans. Currently, there are no specific regulatory guidelines for the fabrication and testing of bioinspired products. Numerous obstacles must be removed before the translating biomimetic nanocarriers from bench to bedside.

## 5. Conclusions

In this work, a novel bioinspired vector was successfully prepared to effectively deliver CUR. The developed PCLP-CUR possessed prolonged circulation time, improved bioavailability, superior tumor targeting capacity and good biocompatibility. Chitosan incorporation could enable PCLP-CUR to facilitate drug release in the acidic environments. Moreover, both in vitro and in vivo assays confirmed that this developed formulation displayed enhanced anti-liver cancer efficacy, which might have been due to the increased uptake, accumulation and action of CUR via platelet membrane camouflage and chitosan modification. In summary, PCLP-CUR is efficient without observable toxicity.

## Data Availability

Data are contained within the article or from the authors upon reasonable request.

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
