# Peer review of "Curcumin-Loaded Platelet Membrane Bioinspired Chitosan-Modified Liposome for Effective Cancer Therapy"

_pharmaceutics, 2023, doi:10.3390/pharmaceutics15020631_

Round 1

Reviewer 1 Report

The authors report a study on the antitumor effects of a curcumin preparation encapsulated in a modified chitosan and liposome platelet membrane system. The study includes standardization of vesicles with curcumin, in vitro and in vivo toxicity studies, pharmacokinetics, distribution, and efficacy.

The study collects a lot of data and multiple results. It seems that the platelet membrane-chitosan-liposomes-curcumin system is effective in reducing the size of the liver tumor in the animal model. However, there are some critical issues that need to be resolved.

1. Regarding the pharmacokinetic data, Figure 3 is unclear. Do the concentrations of ug/L refer to curcumin detected in HPLC? If so, there are no significant differences in pharmacokinetics between i.p. administration of curcumin and curcumin-containing complex (PCLP-CUR). This aspect should be clarified.

2. Another very serious aspect concerns the statistical analysis. The authors report that they applied the t-student. Unfortunately, this statistical test should not be applied to their data. Indeed, they have more than two experimental groups. See, for example, figure 3, but not only.

3. The incorrect statistical approach could question the interpretation of the collected data. I suggest re-analyzing all data with a 1-way or 2-way ANOVA approach when needed.

4. Authors should separate results from discussion. Indeed, they frequently use acronyms that make it difficult to read and given the amount of data it is necessary to have a thorough discussion, especially after a new statistical analysis of the data.

Author Response

Dear Expert Reviewer,

 Thank you very much for the prompt review process and excellent comments. We greatly appreciate the time and efforts which you have spent on it. We are submitting the revised manuscript entitled “Curcumin-loaded Platelet Membrane Bioinspired Chitosan-modified Liposome for Effective Cancer Therapy” (ID: pharmaceutics-2131093) to Pharmaceutics.

We have carefully considered your comments and suggestions, and addressed each of the concerns in response to the comments (see point by point response). We have revised the manuscripts based on your comments and carefully checked throughout the manuscript and improved the language. Our point-by-point responses to the comments (in blue) are shown below (in red).

Comment 1. Regarding the pharmacokinetic data, Figure 3 is unclear. Do the concentrations of ug/L refer to curcumin detected in HPLC? If so, there are no significant differences in pharmacokinetics between i.p. administration of curcumin and curcumin-containing complex (PCLP-CUR). This aspect should be clarified.

Response: Thanks a lot for the constructive and careful suggestion. The concentrations of ug/L refer to curcumin (CUR) was detected in HPLC. And this HPLC method for determination of CUR in rat plasma has been developed and validated in our previous study [28]. Plasma concentration-time profiles of curcumin (Figure 3A) were plotted according to the plasma drug concentration at each time point. And the data were further analyzed by a DAS 3.0 software (Mathematical Pharmacology Professional Committee of China, China), thus the major pharmacokinetic parameters, such as area under concentration-time curve (AUC), mean residence time (MRT), and clearance rate (Cl) were calculated to evaluate the pharmacokinetic behaviors of the formulations (Wang. et al. Int J Pharm 2020; Li. et al. Nanomedicine 2016). The significant difference (P) between each group was analyzed with GraphPad Prism 8 software (GraphPad Software, Inc., USA). One-way ANOVA was utilized for statistical evaluation. P < 0.05 indicated statistical difference. There are significant differences in pharmacokinetic parameters between administration of free CUR and curcumin-containing complex (PCLP-CUR). And we have reanalyzed the data of pharmacokinetics in the result section (page 9, lines 343-353) as follows:

The plasma drug concentration-time curve was shown in Figure 3A. The area under concentration-time curve (AUC) of PCLP-CUR was 336.99 μg/L*h, which was approximately 2.51-fold that of CUR and 1.44-fold that of LP-CUR (Figure 3B). This result indicated that the bioavailability of PCLP-CUR was significantly enhanced compared with that of free CUR and LP-CUR. The mean residence time (MRT) of PCLP-CUR was 3.60 times that of CUR and 2.15 times that of LP-CUR (Figure 3C). While, both CUR and LP-CUR had significantly higher clearance rate (Cl) than PCLP-CUR (Figure 3D). PCLP-CUR with longer MRT and lower Cl values maintained a prolonged circulation time compared to LP-CUR and free CUR.

The figure can be found in the attachment WORD file.

Figure 3. Pharmacokinetics of PCLP-CUR. (A) Plasma drug concentrations versus time profiles after intravenous injection of various formulations in rats (n=5). (B) AUC, (C) MRT, (D) Cl values of CUR, LP-CUR, and PCLP-CUR (n=5). *P < 0.05, **P < 0.01, ***P < 0.001, ****P <0.0001. AUC: area under the plasma concentration-time curve; Cl: clearance rate; MRT: mean residence time.

References

  1. Wan, S.; Wu, Y.; Fan, Q.; Yang, G.; Hu, H.; Tima, S.; Chiampanichayakul, S.; Anuchapreeda, S.; Wu, J. Bioinspired platelet-like nanovector for enhancing cancer therapy via p-selectin targeting. 2022, 14, 2614.

Wang, H.; Luo, J.; Zhang, Y.; He, D.; Jiang, R.; Xie, X.; Yang, Q.; Li, K.; Xie, J.; Zhang, J. Phospholipid/hydroxypropyl-β-cyclodextrin supramolecular complexes are promising candidates for efficient oral delivery of curcuminoids. Int J Pharm 2020, 582, 119301. (Wang. et al. Int J Pharm 2020)

Li, S.; Fang, C.; Zhang, J.; Liu, B.; Wei, Z.; Fan, X.; Sui, Z.; Tan, Q. Catanionic lipid nanosystems improve pharmacokinetics and anti-lung cancer activity of curcumin. Nanomedicine 2016, 12, 1567-1579. (Li. et al. Nanomedicine 2016)

Comment 2. Another very serious aspect concerns the statistical analysis. The authors report that they applied the t-student. Unfortunately, this statistical test should not be applied to their data. Indeed, they have more than two experimental groups. See, for example, figure 3, but not only.

Response: Thank you very much for the valuable advice. We are very sorry for the inappropriate statistical analysis. And we have reanalyzed the data and revised the statistical analysis section (pages 6-7, lines 271-274). The significant difference (P) between each group was analyzed with GraphPad Prism 8 software (GraphPad Software, Inc., USA). One-way ANOVA approach has been applied to the data when there are more than two experimental groups. P < 0.05 indicated statistical difference. According to the data after reanalyzing, the Figure 3 (page 10, lines 354-358), Figure 5A (page 12, lines 406-413), and Figure 5C (page 12, lines 406-413) were redrawn. The revised figures were as follows:

The figure can be found in the attachment WORD file.

Figure 3. Pharmacokinetics of PCLP-CUR. (A) Plasma drug concentrations versus time profiles after intravenous injection of various formulations in rats (n=5). (B) AUC, (C) MRT, (D) Cl values of CUR, LP-CUR, and PCLP-CUR (n=5). *P < 0.05, **P < 0.01, ***P < 0.001, ****P <0.0001. AUC: area under the plasma concentration-time curve; Cl: clearance rate; MRT: mean residence time.

The figure can be found in the attachment WORD file.

Figure 5. Enhanced therapeutic efficacy of PCLP-CUR. (A) Cytotoxicity of various formulations (CUR concentration: 20 µM) against HepG2 cells after 48 h-incubation. **P < 0.01, and ***P < 0.001 (n = 5).

Comment 3. The incorrect statistical approach could question the interpretation of the collected data. I suggest re-analyzing all data with a 1-way or 2-way ANOVA approach when needed.

Response: We gratefully appreciate your valuable suggestion. We are very sorry for the incorrect statistical analysis. And we have reanalyzed the data. In the revised manuscript, the significant difference (P) between each group was analyzed with GraphPad Prism 8 software (GraphPad Software, Inc., USA). One-way ANOVA approach has been applied to the data when there are more than two experimental groups. P < 0.05 indicated statistical difference. According to the data after reanalyzing, the Figure 3 (page 10, lines 354-358), Figure 5A (page 12, lines 406-413), and Figure 5C (page 12, lines 406-413) were redrawn. And we rewrote the result section (page 9, lines 343-353; page 11, lines 387-390; page 11, lines 393-396) as follows:

The plasma drug concentration-time curve was shown in Figure 3A. The area under concentration-time curve (AUC) of PCLP-CUR was 336.99 μg/L*h, which was approximately 2.51-fold that of CUR and 1.44-fold that of LP-CUR (Figure 3B). This result indicated that the bioavailability of PCLP-CUR was significantly enhanced compared with that of free CUR and LP-CUR. The mean residence time (MRT) of PCLP-CUR was 3.60 times that of CUR and 2.15 times that of LP-CUR (Figure 3C). While, both CUR and LP-CUR had significantly higher clearance rate (Cl) than PCLP-CUR (Figure 3D). PCLP-CUR with longer MRT and lower Cl values maintained a prolonged circulation time compared to LP-CUR and free CUR.

The figure can be found in the attachment WORD file.

Figure 3. Pharmacokinetics of PCLP-CUR. (A) Plasma drug concentrations versus time profiles after intravenous injection of various formulations in rats (n=5). (B) AUC, (C) MRT, (D) Cl values of CUR, LP-CUR, and PCLP-CUR (n=5). *P < 0.05, **P < 0.01, ***P < 0.001, ****P <0.0001. AUC: area under the plasma concentration-time curve; Cl: clearance rate; MRT: mean residence time.

As shown in Figure 5A, PCLP-CUR showed a significantly stronger tumor cell killing effect than CUR. Upon chitosan modification, CLP-CUR exhibited higher cytotoxicity than LP-CUR. In addition, compared with CLP-CUR, PCLP-CUR showed a notably higher cytotoxicity after cloaking with platelet membrane.

The figure can be found in the attachment WORD file.

Figure 5. Enhanced therapeutic efficacy of PCLP-CUR. (A) Cytotoxicity of various formulations (CUR concentration: 20 µM) against HepG2 cells after 48 h-incubation. **P < 0.01, and ***P < 0.001 (n = 5).

As shown in Figure 5C, when compared with control group and blank PCLP, both CUR and LP-CUR exhibited moderate effect on the inhibition of tumor growth. While an observably enhanced tumor inhibition effect was found in PCLP-CUR group. The results indicated PCLP-CUR could significantly enhance the anticancer effect of CUR.

The figure can be found in the attachment WORD file.

Figure 5. Enhanced therapeutic efficacy of PCLP-CUR. (C) Tumor growth curves of HepG2 tumor-bearing BALB/c nude mice treated with different formulations. *P < 0.05, ***P < 0.001, ****P <0.0001, (n = 5).

Comment 4. Authors should separate results from discussion. Indeed, they frequently use acronyms that make it difficult to read and given the amount of data it is necessary to have a thorough discussion, especially after a new statistical analysis of the data.

Response: Thank you so much for your scientific review. We have separated results from discussion. And we reduced the frequency of acronyms use in the revised manuscript. After a new statistical analysis of the data, we have improved the result section (pages 7-14, lines 275-440). According to your suggestion, we have made a thorough discussion in the discussion section (pages 14-15, lines 421-493) as follows:

Natural products are a sustainable bioresource with the potential to treat cancer. Many efforts, such as nanotechnology, integrated study approach and network, have been devoted to developing effective natural medicines [4,36]. However, delivery of natural anticancer drugs faces great challenges due to limited in vivo kinetic behaviors and poor tumor accumulation. To improve bioavailability and antitumor effect, we prepared novel biomimetic polysaccharide-cloaked lipidic nanocarriers (PCLP-CUR) to deliver the natural polyphenol CUR. This developed nanoparticle is safe and has high efficacy.

The micromorphology, size, zeta potential, and protein environment suggested the successful fabrication of PCLP-CUR. Chitosan modification ameliorated drug release behavior [37]. According to the drug release rate, the drug release from PCLP-CUR was more rapid in acidic environment than at pH 7.4 (Figure 1G), indicating the pH-responsive drug release ability of PCLP-CUR. The mechanism might be attributed to the protonated amines of chitosan, which can induce the swelling of PCLP-CUR, thus, the CUR molecules could easily diffuse out from the liposomes [15,16]. Due to the anticancer drug release was much faster and greater under acidic conditions than under normal physiological conditions, the side effects of drug could be significantly reduced in normal cells [15,38]. Thus, the newly prepared PCLP-CUR with pH-responsivity could achieve tumor treatment and reduce the adverse effect of therapeutic agents.

The results of cellular uptake demonstrated that PCLP could escape uptake by macrophages (Figure 2A), presumably CD47 protein on the platelet membrane sending “don't eat me” signals to macrophages [39,40]. Phagocytosis of macrophage cells is the primary cause of rapid blood clearance of drug carriers [41]. The prolonged MRT values and decreased Cl suggested the in vivo long circulation of PCLP. The ameliorated in vivo kinetic characteristics should be ascribed to platelet membrane camouflage.

Both in vitro and in vivo assays confirmed that PCLP had a good tumor targeting capacity. As shown in Figure 2B, compared with synthetic liposomes, PCLP was more efficiently taken up by HepG2 cells. And the uptake of PCLP in free HA (a ligand of CD44) preincubated HepG2 cells was reduced. As described in previous reports, P-selectin has a high affinity for the CD44 receptor [31,41]. The enhanced cellular uptake of PCLP was likely attributed to the interaction between the CD44 receptor of HepG2 cell and P-selectin on the platelet membrane. In vivo tumor targeting evaluation confirmed that the accumulation of PCLP in tumor tissues was greater than that of synthetic liposomes, presumably through the platelet membrane cloak [39].

The in vivo anticancer study showed that the tumor size of HepG2 tumor-bearing mice treated with PCLP-CUR was obviously lower than those of the control group and free drug group. The in vivo results of PCLP-CUR were in line with the in vitro antitumor effects. The high anticancer efficiency of PCLP-CUR might be due to the following reasons: the entrapment of CUR inside of the PCLP-CUR improved its in vivo retention time, bioavailability, and tumor specificity, which allowed more drug accumulation at the cancerous site; platelet membrane camouflage contributed to the increasing uptake of CUR by cancer cells through interaction between P-selectin and CD44 [42]; and the modification of chitosan accelerated drug release at the tumor regions [35]. Weight change, organ H&E staining, and serum biochemistry analysis certified that the preliminary safety of PCLP-CUR was good. The cytocompatibility assay also confirmed the high biocompatibility of the developed nanocarrier. This satisfactory safety may be related to the materials used in this study, such as natural polysaccharides and biocompatible cell membranes.

Based on these advantages, PCLP-CUR represents a promising targeted cancer treatment. However, the safety and antitumor efficiency of emerging biomimetic nanoformulations have not been fully demonstrated in humans. Currently, there are no specific regulatory guidelines for fabrication and testing of bioinspired products. Numerous obstacles must be removed before the translating biomimetic nanocarriers from bench to bedside.

References

  1. Singla, R.K.; De, R.; Efferth, T.; Mezzetti, B.; Sahab Uddin, M.; Sanusi; Ntie-Kang, F.; Wang, D.; Schultz, F.; Kharat, K.R., et al. The International Natural Product Sciences Taskforce (INPST) and the power of Twitter networking exemplified through # INPST hashtag analysis. Phytomedicine: international journal of phytotherapy and phytopharmacology 2023, 108, 154520.
  2. Wang, M.; Zhao, T.; Liu, Y.; Wang, Q.; Xing, S.; Li, L.; Wang, L.; Liu, L.; Gao, D. Ursolic acid liposomes with chitosan modification: Promising antitumor drug delivery and efficacy. Mater Sci Eng C Mater Biol Appl 2017, 71, 1231-1240.
  3. Xie, L.; Yang, Y.; Shen, J. Efficient inhibition of uveal melanoma via ternary sirna complexes. Int J Pharm 2020, 573, 118894.
  4. Liu, G.; Zhao, X.; Zhang, Y.; Xu, J.; Xu, J.; Li, Y.; Min, H.; Shi, J.; Zhao, Y.; Wei, J., et al. Engineering biomimetic platesomes for pH-responsive drug delivery and enhanced antitumor activity. Adv Mater 2019, 31, e1900795.
  5. Ren, X.; He, L.; Tian, X.; Zhang, P.; Chen, Z.; Mei, X. pH and folic acid dual responsive polysaccharide nanospheres used for nuclear targeted cancer chemotherapy. Colloids Surf B Biointerfaces 2019, 178, 445-451.
  6. Sachdeva, A.; Dhawan, D.; Jain, G.K.; Yerer, M.B.; Collignon, T.E.; Tewari, D.; Bishayee, A. Novel strategies for the bioavailability augmentation and efficacy improvement of natural products in oral cancer. Cancers (Basel) 2022, 15.
  7. Xie, X.; Li, Y.; Zhao, D.; Fang, C.; He, D.; Yang, Q.; Yang, L.; Chen, R.; Tan, Q.; Zhang, J. Oral administration of natural polyphenol-loaded natural polysaccharide-cloaked lipidic nanocarriers to improve efficacy against small-cell lung cancer. Nanomedicine 2020, 29, 102261.
  8. Cai, X.; Yang, Q.; Weng, Q.; Wang, S. pH sensitive doxorubicin-loaded nanoparticle based on radix pseudostellariae protein-polysaccharide conjugate and its improvement on HepG2 cellular uptake of doxorubicin. Food and chemical toxicology: an international journal published for the British Industrial Biological Research Association 2020, 136, 111099.
  9. Wang, H.; Wu, J.; Williams, G.R.; Fan, Q.; Niu, S.; Wu, J.; Xie, X.; Zhu, L.M. Platelet-membrane-biomimetic nanoparticles for targeted antitumor drug delivery. J Nanobiotechnology 2019, 17, 60.
  10. Xie, W.; Liu, P.; Gao, F.; Gu, Y.; Xiao, Y.; Wu, P.; Chen, B.; Liu, W.; Liu, Q. Platelet-neutrophil hybrid membrane-coated gelatin nanoparticles for enhanced targeting ability and intelligent release in the treatment of non-alcoholic steatohepatitis. Nanomedicine 2022, 42, 102538.
  11. Hu, Q.; Sun, W.; Qian, C.; Wang, C.; Bomba, H.N.; Gu, Z. Anticancer platelet-mimicking nanovehicles. Adv Mater 2015, 27, 7043-7050.
  12. Chen, M.; Qiao, Y.; Cao, J.; Ta, L.; Ci, T.; Ke, X. Biomimetic doxorubicin/ginsenoside co-loading nanosystem for chemoimmunotherapy of acute myeloid leukemia. J Nanobiotechnology 2022, 20, 273.

Thank you for all the valuable and helpful comments and suggestions. We hope that our revised manuscript is now suitable for publication in Pharmaceutics.

Best regards,

Jianming Wu

Reviewer 2 Report

A title that more reflects the content should be given. Curcumin should be mentioned.

The importance  of natural products and nanonutraceuticals in counteract chronic diseases should be marked as well as the importance of an integrated study approach and network and related references should be added such as:

Singla RK, De R, Efferth T, et al. The International Natural Product Sciences Taskforce (INPST) and the power of Twitter networking exemplified through #INPST hashtag analysis. Phytomedicine. 2023;108:154520. doi:10.1016/j.phymed.2022.154520

 Yeung et al.Big impact of nanoparticles: analysis of the most cited nanopharmaceuticals and nanonutraceuticals research,Current Research in Biotechnology,Volume 2,2020,Pages 53-63,ISSN 2590-2628, https://doi.org/10.1016/j.crbiot.2020.04.002.

Main features of curcumin should be described and related references such as:

ZieliÅ„ska, et al Properties, Extraction Methods, and Delivery Systems for Curcumin as a Natural Source of Beneficial Health Effects. Medicina 202056, 336. https://doi.org/10.3390/medicina56070336

The aim should be rewritten and the novelty character of paper should be marked.

Figure 1 G and Figure 1 H should be better described and discussed in the text.

The description of results on  in vitro delivery of PCLP should be implemented.

Results in Figure 4 should be better described and discussed.

Limits, advantages and practical applications

Author Response

Dear Expert Reviewer,

 Thank you very much for the prompt review process and excellent comments. We greatly appreciate the time and efforts which you have spent on it. We are submitting the revised manuscript entitled “Curcumin-loaded Platelet Membrane Bioinspired Chitosan-modified Liposome for Effective Cancer Therapy” (ID: pharmaceutics-2131093) to Pharmaceutics.

We have carefully considered your comments and suggestions, and addressed each of the concerns in response to the comments (see point by point response). We have revised the manuscripts based on your comments and carefully checked throughout the manuscript and improved the language. Our point-by-point responses to the comments (in blue) are shown below (in red).

Comment 1. A title that more reflects the content should be given. Curcumin should be mentioned.

Response: Thank you very much for your precious advice. We have revised the title, and curcumin have been mentioned. The revised title (page 1, lines 2-3) is as follow:

Curcumin-loaded platelet membrane bioinspired chitosan-modified liposome for effective cancer therapy.

Comment 2. The importance of natural products and nanonutraceuticals in counteract chronic diseases should be marked as well as the importance of an integrated study approach and network and related references should be added such as:

Singla RK, De R, Efferth T, et al. The International Natural Product Sciences Taskforce (INPST) and the power of Twitter networking exemplified through #INPST hashtag analysis. Phytomedicine. 2023;108:154520. doi:10.1016/j.phymed.2022.154520

Yeung et al.Big impact of nanoparticles: analysis of the most cited nanopharmaceuticals and nanonutraceuticals research,Current Research in Biotechnology,Volume 2,2020,Pages 53-63,ISSN 2590-2628, https://doi.org/10.1016/j.crbiot.2020.04.002.

Response: Thanks for your kindly attention and consideration. The importance of natural products and nanonutraceuticals in counteract chronic diseases have been marked as well as the importance of an integrated study approach and network in the introduction section (pages 1-2, lines 45-48; page 2, lines 53-54) and discussion section (page 14, lines 443-444). And related references have been added. The revised content is as follows:

Nutraceuticals are intensively investigated in cancer and chronic diseases, and these natural products offer new possibilities for tumor therapies to overcome the shortcomings of traditional chemotherapeutic treatment [3-5]. Nowadays, nanonutraceutical formulations represent a valuable strategy used in managing health conditions. Many efforts, such as nanotechnology, integrated study approach and network, have been devoted to developing effective natural medicines [4,36].

References

  1. Yeung, A.W.K.; Souto, E.B.; Durazzo, A.; Lucarini, M.; Novellino, E.; Tewari, D.; Wang, D.; Atanasov, A.G.; Santini, A. Big impact of nanoparticles: Analysis of the most cited nanopharmaceuticals and nanonutraceuticals research. Current Research in Biotechnology 2020, 2, 53-63.
  2. Singla, R.K.; De, R.; Efferth, T.; Mezzetti, B.; Sahab Uddin, M.; Sanusi; Ntie-Kang, F.; Wang, D.; Schultz, F.; Kharat, K.R., et al. The International Natural Product Sciences Taskforce (INPST) and the power of twitter networking exemplified through # INPST hashtag analysis. Phytomedicine: international journal of phytotherapy and phytopharmacology 2023, 108, 154520.
  3. Li, Y.; Jia, Y.; Wang, X.; Shang, H.; Tian, Y. Protein-targeted degradation agents based on natural products. Pharmaceuticals (Basel, Switzerland) 2022, 16.
  4. Sachdeva, A.; Dhawan, D.; Jain, G.K.; Yerer, M.B.; Collignon, T.E.; Tewari, D.; Bishayee, A. Novel strategies for the bioavailability augmentation and efficacy improvement of natural products in oral cancer. Cancers (Basel) 2022, 15.

Comment 3. Main features of curcumin should be described and related references such as:

Zielińska, et al Properties, Extraction Methods, and Delivery Systems for Curcumin as a Natural Source of Beneficial Health Effects. Medicina 2020, 56, 336. https://doi.org/10.3390/medicina56070336

Response: Thank you very much for your valuable advice. Main features of curcumin have been described in the introduction section (page 2, lines 48-52). And related references have been added. The revised content is as follows:

Curcumin (CUR), a curcuminoid, extracted from the rhizomes of Curcuma longa, has several advantages, such as strong anticancer effect and outstanding safety [6-9]. Nevertheless, CUR is a hydrophobic molecule with low bioavailability [10]. Meanwhile, rapid blood clearance and limited tumor accumulation have hindered the clinical application of natural compounds including CUR.

References

  1. Soni, V.K.; Mehta, A.; Ratre, Y.K.; Chandra, V.; Shukla, D.; Kumar, A.; Vishvakarma, N.K. Counteracting action of curcumin on high glucose-induced chemoresistance in hepatic carcinoma cells. Front Oncol 2021, 11, 738961.
  2. Wang, Y.; Ding, R.; Zhang, Z.; Zhong, C.; Wang, J.; Wang, M. Curcumin-loaded liposomes with the hepatic and lysosomal dual-targeted effects for therapy of hepatocellular carcinoma. Int J Pharm 2021, 602, 120628.
  3. Rawat, D.; Shrivastava, S.; Naik, R.A.; Chhonker, S.K.; Mehrotra, A.; Koiri, R.K. An overview of natural plant products in the treatment of hepatocellular carcinoma. Anticancer Agents Med Chem 2018, 18, 1838-1859.
  4. Hu, Y.; Wang, S.; Wu, X.; Zhang, J.; Chen, R.; Chen, M.; Wang, Y. Chinese herbal medicine-derived compounds for cancer therapy: A focus on hepatocellular carcinoma. J Ethnopharmacol 2013, 149, 601-612.
  5. Zielińska, A.; Alves, H.; Marques, V.; Durazzo, A.; Lucarini, M.; Alves, T.F.; Morsink, M.; Willemen, N.; Eder, P.; Chaud, M.V., et al. Properties, extraction methods, and delivery systems for curcumin as a natural source of beneficial health effects. Medicina (Kaunas, Lithuania) 2020, 56.

Comment 4. The aim should be rewritten and the novelty character of paper should be marked.

Response: We gratefully appreciate your precious advice. We have rewritten the aim and marked the novelty character of paper (page 2, lines 82-86) as follows:

To date, reports on the delivery of natural antitumor drugs by platelet membrane bio-mimetic nanoparticles are still limited. In our work, a novel CUR-loaded platelet membrane bioinspired chitosan-modified liposome (PCLP-CUR) was developed to improve the release, pharmacokinetic characteristics, tumor targeting, and anticancer effect of CUR.

Comment 5. Figure 1 G and Figure 1 H should be better described and discussed in the text.

Response: Thank you so much for your scientific review. In the revised manuscript, we have separated results from discussion and made a thorough discussion in the discussion section. The describe (page 7, lines 301-309) and discussion (page 14, lines 451-459) of Figure 1 G and Figure 1 H have been improved in the revised manuscript as follows:

Drug release from PCLP-CUR or PLP-CUR was measured at pH 7.4 (simulating the normal physiological environment), pH 6.5, and pH 5.0 (simulating the acidic micro-environment of the tumor) [35]. A typical biphasic pattern with an initial quick release phase followed by a slow sustained release phase was observed in both the PCLP-CUR group and PLP-CUR group (Figure 1G and Figure 1H). After dialysis for 24 h, the release rates of CUR from PCLP-CUR were approximately 60% at pH 5.0, 49% at pH 6.5, and only 34% at pH 7.4 (Figure 1G). The release of CUR from PCLP-CUR was much faster in the acidic environment than that at pH 7.4. In contrast, the drug release profiles of PLP-CUR, the formulation without chitosan modification, were similar at different pH conditions (Figure 1H).

According to the drug release rate, the drug release from PCLP-CUR was more rapid in acidic environment than at pH 7.4 (Figure 1G), indicating the pH-responsive drug release ability of PCLP-CUR. The mechanism might be attributed to the protonated amines of chitosan, which can induce the swelling of PCLP-CUR, thus, the CUR molecules could easily diffuse out from the liposomes [15,16]. Due to the anticancer drug release was much faster and greater under acidic conditions than under normal physiological conditions, the side effects of drug could be significantly reduced in normal cells [15,38]. Thus, the newly prepared PCLP-CUR with pH-responsivity could achieve tumor treatment and reduce the adverse effect of therapeutic agents.

References

  1. Wang, M.; Zhao, T.; Liu, Y.; Wang, Q.; Xing, S.; Li, L.; Wang, L.; Liu, L.; Gao, D. Ursolic acid liposomes with chitosan modification: Promising antitumor drug delivery and efficacy. Mater Sci Eng C Mater Biol Appl 2017, 71, 1231-1240.
  2. Xie, L.; Yang, Y.; Shen, J. Efficient inhibition of uveal melanoma via ternary sirna complexes. Int J Pharm 2020, 573, 118894.
  3. Ren, X.; He, L.; Tian, X.; Zhang, P.; Chen, Z.; Mei, X. Ph and folic acid dual responsive polysaccharide nanospheres used for nuclear targeted cancer chemotherapy. Colloids Surf B Biointerfaces 2019, 178, 445-451.
  4. Cai, X.; Yang, Q.; Weng, Q.; Wang, S. pH sensitive doxorubicin-loaded nanoparticle based on radix pseudostellariae protein-polysaccharide conjugate and its improvement on HepG2 cellular uptake of doxorubicin. Food and chemical toxicology: an international journal published for the British Industrial Biological Research Association 2020, 136, 111099.

Comment 6. The description of results on in vitro delivery of PCLP should be implemented.

Response: We gratefully appreciate your valuable suggestion. The description of results on in vitro delivery of PCLP have been implemented in the manuscript (pages 8-9, lines 320-336) as follows:

Cell membrane biomimetic nanoparticles can prolong the in vivo circulation time through immune escape [24]. To demonstrate the immune escape ability of PCLP, the phagocytosis of RAW264.7 macrophages was evaluated. RAW264.7 cells treated with LP-DiD or CLP-DiD showed higher red fluorescence intensities than those treated with PCLP-DiD (Figure 2A), suggesting the prepared bioinspired liposome could escape uptake by macrophages. The uptake of PCLP-DiD in CD44 overexpressing HepG2 cells was also examined. The fluorescence signals in HepG2 cells treated with PCLP-DiD were stronger than those of the synthetic liposome groups (Figure 2B), indicating that PCLP-DiD was more efficiently taken up by HepG2 cells. These results confirmed that PCLP could successfully escape uptake by macrophages and drive cargo into HepG2 cells efficiently. To further explore the endocytosis mechanism of cellular uptake, an experiment of competitive inhibition was performed in which HepG2 cells were pretreated with free HA. HA is a strong-affinity ligand of the CD44 receptor [34]. Confocal laser scanning microscope images (Figure 2B) showed that the fluorescence intensity of PCLP in free HA preincubated HepG2 cells was reduced. This result indicated that the CD44 receptor was involved in the recognition process of the developed biomimetic liposome and HepG2 cells.

References

  1. Ye, H.; Wang, K.; Wang, M.; Liu, R.; Song, H.; Li, N.; Lu, Q.; Zhang, W.; Du, Y.; Yang, W., et al. Bioinspired nanoplatelets for chemo-photothermal therapy of breast cancer metastasis inhibition. Biomaterials 2019, 206, 1-12.

Comment 7. Results in Figure 4 should be better described and discussed.

Response: Thank you very much for your precious advice. In the revised manuscript, we have separated results from discussion and made a thorough discussion in the discussion section. The describe (page 10, lines 360-370) and discussion (page 14, lines 472-474) of Figure 4 have been improved in the revised manuscript as follows:

The biodistribution of PCLP in HepG2 tumor bearing nude mice was investigated. As shown in Figure 4A, at 24 h after intravenous administration, LP-DiR was found to moderately concentrate at the tumor sites, which was likely attributed to the EPR effect of liposome. In contrast, the fluorescence intensity of PCLP-DiR in tumor tissues was higher and lasted longer than that of the LP-DiR group, suggesting that the developed bioinspired liposome had a superior targeting capacity. The ex vivo images of excised organs at 24 h after treatment also confirmed this result. As shown in Figures 4B and 4C, a higher DiR signal was observed in the tumor sites of PCLP-DiR group than that of LP-DiR group. These results were in good accordance with the in vitro cellular uptake. The sustained high-dose retention in tumor tissues makes PCLP a promising delivery system for anticancer agents. In vivo tumor targeting evaluation confirmed that the accumulation of PCLP in tumor tissues was greater than that of synthetic liposomes, presumably through the platelet membrane cloak [39].

References

  1. Wang, H.; Wu, J.; Williams, G.R.; Fan, Q.; Niu, S.; Wu, J.; Xie, X.; Zhu, L.M. Platelet-membrane-biomimetic nanoparticles for targeted antitumor drug delivery. J Nanobiotechnology 2019, 17, 60.

Comment 8. Limits, advantages and practical applications.

Response: We gratefully appreciate your valuable suggestion. We have added the limits, advantages and practical applications of PCLP-CUR in discussion section (pages 14-15, lines 441-493) as follows:

Natural products are a sustainable bioresource with the potential to treat cancer. Many efforts, such as nanotechnology, integrated study approach and network, have been devoted to developing effective natural medicines [4,36]. However, delivery of natural anticancer drugs faces great challenges due to limited in vivo kinetic behaviors and poor tumor accumulation. To improve bioavailability and antitumor effect, we prepared novel biomimetic polysaccharide-cloaked lipidic nanocarriers (PCLP-CUR) to deliver the natural polyphenol CUR. This developed nanoparticle is safe and has high efficacy.

The micromorphology, size, zeta potential, and protein environment suggested the successful fabrication of PCLP-CUR. Chitosan modification ameliorated drug release behavior [37]. According to the drug release rate, the drug release from PCLP-CUR was more rapid in acidic environment than at pH 7.4 (Figure 1G), indicating the pH-responsive drug release ability of PCLP-CUR. The mechanism might be attributed to the protonated amines of chitosan, which can induce the swelling of PCLP-CUR, thus, the CUR molecules could easily diffuse out from the liposomes [15,16]. Due to the anticancer drug release was much faster and greater under acidic conditions than under normal physiological conditions, the side effects of drug could be significantly reduced in normal cells [15,38]. Thus, the newly prepared PCLP-CUR with pH-responsivity could achieve tumor treatment and reduce the adverse effect of therapeutic agents.

The results of cellular uptake demonstrated that PCLP could escape uptake by macrophages (Figure 2A), presumably CD47 protein on the platelet membrane sending “don't eat me” signals to macrophages [39,40]. Phagocytosis of macrophage cells is the primary cause of rapid blood clearance of drug carriers [41]. The prolonged MRT values and decreased Cl suggested the in vivo long circulation of PCLP. The ameliorated in vivo kinetic characteristics should be ascribed to platelet membrane camouflage.

Both in vitro and in vivo assays confirmed that PCLP had a good tumor targeting capacity. As shown in Figure 2B, compared with synthetic liposomes, PCLP was more efficiently taken up by HepG2 cells. And the uptake of PCLP in free HA (a ligand of CD44) preincubated HepG2 cells was reduced. As described in previous reports, P-selectin has a high affinity for the CD44 receptor [31,41]. The enhanced cellular uptake of PCLP was likely attributed to the interaction between the CD44 receptor of HepG2 cell and P-selectin on the platelet membrane. In vivo tumor targeting evaluation confirmed that the accumulation of PCLP in tumor tissues was greater than that of synthetic liposomes, presumably through the platelet membrane cloak [39].

The in vivo anticancer study showed that the tumor size of HepG2 tumor-bearing mice treated with PCLP-CUR was obviously lower than those of the control group and free drug group. The in vivo results of PCLP-CUR were in line with the in vitro antitumor effects. The high anticancer efficiency of PCLP-CUR might be due to the following reasons: the entrapment of CUR inside of the PCLP-CUR improved its in vivo retention time, bioavailability, and tumor specificity, which allowed more drug accumulation at the cancerous site; platelet membrane camouflage contributed to the increasing uptake of CUR by cancer cells through interaction between P-selectin and CD44 [42]; and the modification of chitosan accelerated drug release at the tumor regions [35]. Weight change, organ H&E staining, and serum biochemistry analysis certified that the preliminary safety of PCLP-CUR was good. The cytocompatibility assay also confirmed the high biocompatibility of the developed nanocarrier. This satisfactory safety may be related to the materials used in this study, such as natural polysaccharides and biocompatible cell membranes.

Based on these advantages, PCLP-CUR represents a promising targeted cancer treatment. However, the safety and antitumor efficiency of emerging biomimetic nanoformulations have not been fully demonstrated in humans. Currently, there are no specific regulatory guidelines for fabrication and testing of bioinspired products. Numerous obstacles must be removed before the translating biomimetic nanocarriers from bench to bedside.

References

  1. Singla, R.K.; De, R.; Efferth, T.; Mezzetti, B.; Sahab Uddin, M.; Sanusi; Ntie-Kang, F.; Wang, D.; Schultz, F.; Kharat, K.R., et al. The International Natural Product Sciences Taskforce (INPST) and the power of Twitter networking exemplified through # INPST hashtag analysis. Phytomedicine: international journal of phytotherapy and phytopharmacology 2023, 108, 154520.
  2. Wang, M.; Zhao, T.; Liu, Y.; Wang, Q.; Xing, S.; Li, L.; Wang, L.; Liu, L.; Gao, D. Ursolic acid liposomes with chitosan modification: Promising antitumor drug delivery and efficacy. Mater Sci Eng C Mater Biol Appl 2017, 71, 1231-1240.
  3. Xie, L.; Yang, Y.; Shen, J. Efficient inhibition of uveal melanoma via ternary sirna complexes. Int J Pharm 2020, 573, 118894.
  4. Liu, G.; Zhao, X.; Zhang, Y.; Xu, J.; Xu, J.; Li, Y.; Min, H.; Shi, J.; Zhao, Y.; Wei, J., et al. Engineering biomimetic platesomes for pH-responsive drug delivery and enhanced antitumor activity. Adv Mater 2019, 31, e1900795.
  5. Ren, X.; He, L.; Tian, X.; Zhang, P.; Chen, Z.; Mei, X. pH and folic acid dual responsive polysaccharide nanospheres used for nuclear targeted cancer chemotherapy. Colloids Surf B Biointerfaces 2019, 178, 445-451.
  6. Sachdeva, A.; Dhawan, D.; Jain, G.K.; Yerer, M.B.; Collignon, T.E.; Tewari, D.; Bishayee, A. Novel strategies for the bioavailability augmentation and efficacy improvement of natural products in oral cancer. Cancers (Basel) 2022, 15.
  7. Xie, X.; Li, Y.; Zhao, D.; Fang, C.; He, D.; Yang, Q.; Yang, L.; Chen, R.; Tan, Q.; Zhang, J. Oral administration of natural polyphenol-loaded natural polysaccharide-cloaked lipidic nanocarriers to improve efficacy against small-cell lung cancer. Nanomedicine 2020, 29, 102261.
  8. Cai, X.; Yang, Q.; Weng, Q.; Wang, S. pH sensitive doxorubicin-loaded nanoparticle based on radix pseudostellariae protein-polysaccharide conjugate and its improvement on HepG2 cellular uptake of doxorubicin. Food and chemical toxicology: an international journal published for the British Industrial Biological Research Association 2020, 136, 111099.
  9. Wang, H.; Wu, J.; Williams, G.R.; Fan, Q.; Niu, S.; Wu, J.; Xie, X.; Zhu, L.M. Platelet-membrane-biomimetic nanoparticles for targeted antitumor drug delivery. J Nanobiotechnology 2019, 17, 60.
  10. Xie, W.; Liu, P.; Gao, F.; Gu, Y.; Xiao, Y.; Wu, P.; Chen, B.; Liu, W.; Liu, Q. Platelet-neutrophil hybrid membrane-coated gelatin nanoparticles for enhanced targeting ability and intelligent release in the treatment of non-alcoholic steatohepatitis. Nanomedicine 2022, 42, 102538.
  11. Hu, Q.; Sun, W.; Qian, C.; Wang, C.; Bomba, H.N.; Gu, Z. Anticancer platelet-mimicking nanovehicles. Adv Mater 2015, 27, 7043-7050.
  12. Chen, M.; Qiao, Y.; Cao, J.; Ta, L.; Ci, T.; Ke, X. Biomimetic doxorubicin/ginsenoside co-loading nanosystem for chemoimmunotherapy of acute myeloid leukemia. J Nanobiotechnology 2022, 20, 273.

Thank you for all the valuable and helpful comments and suggestions. We hope that our revised manuscript is now suitable for publication in Pharmaceutics.

Best regards,

Jianming Wu

Reviewer 3 Report

The present work is of interest, and the results presented seems to be of special interest for future applications.

The paper is well prepared and well written, I just regret the lack of some details as about:

- The figure legends need to be more complete. Example in Figure 1A add information about the type of microscopy (TEM here) and in Figure 1D (confocal microscopy).

- References 24 and 26 are not for curcumin-loaded platelet membrane bioinspired liposom fabrication. Could you please more clearly specify the quantity of curcumin here used to prepare each type of bioinspired liposom?

- Discussion of the results. Only 34 references most of them for the introduction and experimental sections. Please include more detailled discussion about the mechanism, advantage, selectivity, and safety.

Author Response

Dear Expert Reviewer,

 Thank you very much for the prompt review process and excellent comments. We greatly appreciate the time and efforts which you have spent on it. We are submitting the revised manuscript entitled “Curcumin-loaded Platelet Membrane Bioinspired Chitosan-modified Liposome for Effective Cancer Therapy” (ID: pharmaceutics-2131093) to Pharmaceutics.

We have carefully considered your comments and suggestions, and addressed each of the concerns in response to the comments (see point by point response). We have revised the manuscripts based on your comments and carefully checked throughout the manuscript and improved the language. Our point-by-point responses to the comments (in blue) are shown below (in red).

Comment 1. The figure legends need to be more complete. Example in Figure 1A add information about the type of microscopy (TEM here) and in Figure 1D (confocal microscopy).

Response: Thank you very much for your precious advice. We have improved the figure legends. And we have added the information about the instruments (page 8, lines 311-318; page 9, lines 338-342; page 11, lines 380-384), such as the type of microscopy, the type of confocal microscopy, and the type of in vivo image system. The revised content is as follows:

Figure 1. Characterization of PCLP-CUR. (A) Morphologies observed by transmission electron microscopy (upper panel) and size distributions observed by a Zetasizer (bottom panel). (C) Zeta potentials of various nanoparticles observed by a Zetasizer. (D) Fluorescence images of PCLP observed by confocal laser scanning microscopy.

Figure 2. Efficient in vitro delivery of PCLP observed by confocal laser scanning microscopy.

Figure 4. In vivo tumor targeting capability of PCLP observed by a multimodal animal imaging system.

Comment 2. References 24 and 26 are not for curcumin-loaded platelet membrane bioinspired liposom fabrication. Could you please more clearly specify the quantity of curcumin here used to prepare each type of bioinspired liposom?

Response: We gratefully appreciate your valuable suggestion. 8 mg of curcumin was used to prepare each type of bioinspired liposome. The preparation of curcumin-loaded platelet membrane bioinspired liposome was described in our previous report [28]. And the details of bioinspired liposomes preparations are listed as follows:

For fabrication of CUR-loaded platelet membrane bioinspired chitosan-modified liposome (PCLP-CUR), CUR-loaded chitosan-modified liposome (CLP-CUR) was firstly prepared, followed by cloaking with platelet membrane to form PCLP-CUR. The CLP-CUR was prepared by our previously reported thin film dispersion method with slight modifications [28,30]. In brief, egg yolk lecithin, cholesterol, TPGS, and CUR (8 mg) were dissolved in 30 mL of dichloromethane. The solvent was evaporated to form a dry lipid film. Next, the film was hydrated for 2 h with 10 mL of PBS to form an aqueous suspension, thus CUR-loaded liposome (LP-CUR) was prepared. And then chitosan solution (0.2%, weight/volume) was further dropwise added to this suspension under magnetic stirring for 60 min at 25 °C to prepare CLP-CUR. To prepare PCLP-CUR, the platelet membrane was incubated with CLP-CUR for 30 min, the resultant suspensions were sonicated for 5 min, and then extruded sequentially through polycarbonate membranes with an Avanti Polar Lipids mini extruder [31-33].

CUR-loaded platelet membrane biomimetic liposome (PLP-CUR) was prepared by the same method without the addition of chitosan. Firstly, LP-CUR was prepared by a thin film dispersion method. In brief, egg yolk lecithin, cholesterol, TPGS, and CUR (8 mg) were dissolved in 30 mL of dichloromethane. The solvent was evaporated to form a dry lipid film. Next, the film was hydrated for 2 h with 10 mL of PBS to form an aqueous suspension, thus LP-CUR was prepared [28]. To prepare PLP-CUR, the platelet membrane was incubated LP-CUR with for 30 min, the resultant suspensions were sonicated for 5 min, and then extruded sequentially through polycarbonate membranes with an Avanti Polar Lipids mini extruder [31-33].

References

  1. Wan, S.; Wu, Y.; Fan, Q.; Yang, G.; Hu, H.; Tima, S.; Chiampanichayakul, S.; Anuchapreeda, S.; Wu, J. Bioinspired platelet-like nanovector for enhancing cancer therapy via p-selectin targeting. 2022, 14, 2614.
  2. Liu, G.; Zhao, X.; Zhang, Y.; Xu, J.; Xu, J.; Li, Y.; Min, H.; Shi, J.; Zhao, Y.; Wei, J., et al. Engineering biomimetic platesomes for pH-responsive drug delivery and enhanced antitumor activity. Adv Mater 2019, 31, e1900795.
  3. Bang, K.H.; Na, Y.G.; Huh, H.W.; Hwang, S.J.; Kim, M.S.; Kim, M.; Lee, H.K.; Cho, C.W. The delivery strategy of paclitaxel nanostructured lipid carrier coated with platelet membrane. Cancers (Basel) 2019, 11, 807.
  4. Zhang, W.; Gong, C.; Chen, Z.; Li, M.; Li, Y.; Gao, J. Tumor microenvironment-activated cancer cell membrane-liposome hybrid nanoparticle-mediated synergistic metabolic therapy and chemotherapy for non-small cell lung cancer. J Nanobiotechnology 2021, 19, 339.

Comment 3. Discussion of the results. Only 34 references most of them for the introduction and experimental sections. Please include more detailled discussion about the mechanism, advantage, selectivity, and safety.

Response: Thank you very much for your valuable advice. We are very sorry for the poor discussion. In the revised manuscript, we have separated results from discussion and made a thorough discussion in the discussion section (pages 14-15, lines 441-493). And the related references were also added. The revised content is as follows:

Natural products are a sustainable bioresource with the potential to treat cancer. Many efforts, such as nanotechnology, integrated study approach and network, have been devoted to developing effective natural medicines [4,36]. However, delivery of natural anticancer drugs faces great challenges due to limited in vivo kinetic behaviors and poor tumor accumulation. To improve bioavailability and antitumor effect, we prepared novel biomimetic polysaccharide-cloaked lipidic nanocarriers (PCLP-CUR) to deliver the natural polyphenol CUR. This developed nanoparticle is safe and has high efficacy.

The micromorphology, size, zeta potential, and protein environment suggested the successful fabrication of PCLP-CUR. Chitosan modification ameliorated drug release behavior [37]. According to the drug release rate, the drug release from PCLP-CUR was more rapid in acidic environment than at pH 7.4 (Figure 1G), indicating the pH-responsive drug release ability of PCLP-CUR. The mechanism might be attributed to the protonated amines of chitosan, which can induce the swelling of PCLP-CUR, thus, the CUR molecules could easily diffuse out from the liposomes [15,16]. Due to the anticancer drug release was much faster and greater under acidic conditions than under normal physiological conditions, the side effects of drug could be significantly reduced in normal cells [15,38]. Thus, the newly prepared PCLP-CUR with pH-responsivity could achieve tumor treatment and reduce the adverse effect of therapeutic agents.

The results of cellular uptake demonstrated that PCLP could escape uptake by macrophages (Figure 2A), presumably CD47 protein on the platelet membrane sending “don't eat me” signals to macrophages [39,40]. Phagocytosis of macrophage cells is the primary cause of rapid blood clearance of drug carriers [41]. The prolonged MRT values and decreased Cl suggested the in vivo long circulation of PCLP. The ameliorated in vivo kinetic characteristics should be ascribed to platelet membrane camouflage.

Both in vitro and in vivo assays confirmed that PCLP had a good tumor targeting capacity. As shown in Figure 2B, compared with synthetic liposomes, PCLP was more efficiently taken up by HepG2 cells. And the uptake of PCLP in free HA (a ligand of CD44) preincubated HepG2 cells was reduced. As described in previous reports, P-selectin has a high affinity for the CD44 receptor [31,41]. The enhanced cellular uptake of PCLP was likely attributed to the interaction between the CD44 receptor of HepG2 cell and P-selectin on the platelet membrane. In vivo tumor targeting evaluation confirmed that the accumulation of PCLP in tumor tissues was greater than that of synthetic liposomes, presumably through the platelet membrane cloak [39].

The in vivo anticancer study showed that the tumor size of HepG2 tumor-bearing mice treated with PCLP-CUR was obviously lower than those of the control group and free drug group. The in vivo results of PCLP-CUR were in line with the in vitro antitumor effects. The high anticancer efficiency of PCLP-CUR might be due to the following reasons: the entrapment of CUR inside of the PCLP-CUR improved its in vivo retention time, bioavailability, and tumor specificity, which allowed more drug accumulation at the cancerous site; platelet membrane camouflage contributed to the increasing uptake of CUR by cancer cells through interaction between P-selectin and CD44 [42]; and the modification of chitosan accelerated drug release at the tumor regions [35]. Weight change, organ H&E staining, and serum biochemistry analysis certified that the preliminary safety of PCLP-CUR was good. The cytocompatibility assay also confirmed the high biocompatibility of the developed nanocarrier. This satisfactory safety may be related to the materials used in this study, such as natural polysaccharides and biocompatible cell membranes.

Based on these advantages, PCLP-CUR represents a promising targeted cancer treatment. However, the safety and antitumor efficiency of emerging biomimetic nanoformulations have not been fully demonstrated in humans. Currently, there are no specific regulatory guidelines for fabrication and testing of bioinspired products. Numerous obstacles must be removed before the translating biomimetic nanocarriers from bench to bedside.

References

  1. Singla, R.K.; De, R.; Efferth, T.; Mezzetti, B.; Sahab Uddin, M.; Sanusi; Ntie-Kang, F.; Wang, D.; Schultz, F.; Kharat, K.R., et al. The International Natural Product Sciences Taskforce (INPST) and the power of Twitter networking exemplified through # INPST hashtag analysis. Phytomedicine: international journal of phytotherapy and phytopharmacology 2023, 108, 154520.
  2. Wang, M.; Zhao, T.; Liu, Y.; Wang, Q.; Xing, S.; Li, L.; Wang, L.; Liu, L.; Gao, D. Ursolic acid liposomes with chitosan modification: Promising antitumor drug delivery and efficacy. Mater Sci Eng C Mater Biol Appl 2017, 71, 1231-1240.
  3. Xie, L.; Yang, Y.; Shen, J. Efficient inhibition of uveal melanoma via ternary sirna complexes. Int J Pharm 2020, 573, 118894.
  4. Liu, G.; Zhao, X.; Zhang, Y.; Xu, J.; Xu, J.; Li, Y.; Min, H.; Shi, J.; Zhao, Y.; Wei, J., et al. Engineering biomimetic platesomes for pH-responsive drug delivery and enhanced antitumor activity. Adv Mater 2019, 31, e1900795.
  5. Ren, X.; He, L.; Tian, X.; Zhang, P.; Chen, Z.; Mei, X. pH and folic acid dual responsive polysaccharide nanospheres used for nuclear targeted cancer chemotherapy. Colloids Surf B Biointerfaces 2019, 178, 445-451.
  6. Sachdeva, A.; Dhawan, D.; Jain, G.K.; Yerer, M.B.; Collignon, T.E.; Tewari, D.; Bishayee, A. Novel strategies for the bioavailability augmentation and efficacy improvement of natural products in oral cancer. Cancers (Basel) 2022, 15.
  7. Xie, X.; Li, Y.; Zhao, D.; Fang, C.; He, D.; Yang, Q.; Yang, L.; Chen, R.; Tan, Q.; Zhang, J. Oral administration of natural polyphenol-loaded natural polysaccharide-cloaked lipidic nanocarriers to improve efficacy against small-cell lung cancer. Nanomedicine 2020, 29, 102261.
  8. Cai, X.; Yang, Q.; Weng, Q.; Wang, S. pH sensitive doxorubicin-loaded nanoparticle based on radix pseudostellariae protein-polysaccharide conjugate and its improvement on HepG2 cellular uptake of doxorubicin. Food and chemical toxicology: an international journal published for the British Industrial Biological Research Association 2020, 136, 111099.
  9. Wang, H.; Wu, J.; Williams, G.R.; Fan, Q.; Niu, S.; Wu, J.; Xie, X.; Zhu, L.M. Platelet-membrane-biomimetic nanoparticles for targeted antitumor drug delivery. J Nanobiotechnology 2019, 17, 60.
  10. Xie, W.; Liu, P.; Gao, F.; Gu, Y.; Xiao, Y.; Wu, P.; Chen, B.; Liu, W.; Liu, Q. Platelet-neutrophil hybrid membrane-coated gelatin nanoparticles for enhanced targeting ability and intelligent release in the treatment of non-alcoholic steatohepatitis. Nanomedicine 2022, 42, 102538.
  11. Hu, Q.; Sun, W.; Qian, C.; Wang, C.; Bomba, H.N.; Gu, Z. Anticancer platelet-mimicking nanovehicles. Adv Mater 2015, 27, 7043-7050.
  12. Chen, M.; Qiao, Y.; Cao, J.; Ta, L.; Ci, T.; Ke, X. Biomimetic doxorubicin/ginsenoside co-loading nanosystem for chemoimmunotherapy of acute myeloid leukemia. J Nanobiotechnology 2022, 20, 273.

Thank you for all the valuable and helpful comments and suggestions. We hope that our revised manuscript is now suitable for publication in Pharmaceutics.

Best regards,

Jianming Wu

Round 2

Reviewer 1 Report

The authors corrected and improved their manuscript following the indications of the reviewers. Now in my opinion the manuscript can be published.